# The Impact of *BPI* Expression on *Escherichia coli* F18 Infection in Porcine Kidney Cells

**DOI:** 10.3390/ani10112118

**Published:** 2020-11-15

**Authors:** Jian Jin, Yanjie Huang, Shouyong Sun, Zhengchang Wu, Shenglong Wu, Zongjun Yin, Wenbin Bao

**Affiliations:** 1Key Laboratory for Animal Genetics, Breeding, Reproduction and Molecular Design, College of Animal Science and Technology, Yangzhou University, Yangzhou 225009, China; jianj1127@163.com (J.J.); m18252719706@163.com (Y.H.); dkxy@yzu.edu.cn (S.S.); zcwu@yzu.edu.cn (Z.W.); slwu@yzu.edu.cn (S.W.); 2Joint International Research Laboratory of Agriculture & Agri-Product Safety, Yangzhou University, Yangzhou 225009, China; 3College of Animal Science and Technology, Anhui Agricultural University, Hefei 230036, China; yinzongjun@ahau.edu.cn

**Keywords:** pigs, *BPI* gene, overexpression, *Escherichia coli* F18, antibacterial, cell adhesion

## Abstract

**Simple Summary:**

*Escherichia coli* frequently causes bacterial diarrhea in piglets. Vaccine development and improved feeding and animal management strategies have reduced the incidence of bacterial diarrhea in piglets to some extent. However, current breeding strategies also have the potential to improve piglet resistance to diarrhea at a genetic level. This study sought to advance the current understanding of the functional and regulatory mechanisms whereby the candidate gene bactericidal/permeability-increasing protein (*BPI*) regulates piglet diarrhea at the cellular level.

**Abstract:**

The efficacy and regulatory activity of bactericidal/permeability-increasing protein (*BPI*) as a mediator of *Escherichia coli* (*E. coli*) F18 resistance remains to be defined. In the present study, we evaluated lipopolysaccharide (LPS)-induced changes in *BPI* gene expression in porcine kidney (PK15) cells in response to *E. coli* F18 exposure. We additionally generated PK15 cells that overexpressed *BPI* to assess the impact of this gene on Toll-like receptor 4 (TLR4) signaling and glycosphingolipid biosynthesis-related genes. Through these analyses, we found that *BPI* expression rose significantly following LPS exposure in response to *E. coli* F18ac stimulation (*p* < 0.01). Colony count assays and qPCR analyses revealed that *E. coli* F18 adherence to PK15 cells was markedly suppressed following *BPI* overexpression (*p* < 0.01). *BPI* overexpression had no significant effect on the mRNA-level expression of genes associated with glycosphingolipid biosynthesis or TLR4 signaling. *BPI* overexpression suppressed the LPS-induced TLR4 signaling pathway-related expression of proinflammatory cytokines (*IFN-α*, *IFN-β*, *MIP-1α*, *MIP-1β* and *IL-6*). Overall, our study serves as an overview of the association between *BPI* and resistance to *E. coli* F18 at the cellular level, offering a framework for future investigations of the mechanisms whereby piglets are able to resist *E. coli* F18 infection.

## 1. Introduction

Bactericidal/permeability-increasing protein (*BPI*) is expressed at high levels in a range of animal cell and tissue types, with particularly pronounced expression being evident in neutrophils [1,2]. *BPI* exerts a range of antibacterial functions that enable it to protect against certain diseases in humans by neutralizing lipopolysaccharide (LPS) and killing Gram-negative bacteria [3]. Fan et al. [4] previously demonstrated a relationship between *BPI* polymorphisms and the incidence of ulcerative colitis in humans, while serum *BPI* levels have been utilized as a metric to analyze lipid metabolism and the functionality of the vascular endothelium in humans [5]. *BPI* can also prevent angiogenesis and drive the apoptotic death of endothelial cells, thereby influencing the treatment of a wide range of diseases including arthritis, atherosclerosis and cancer [6]. There is evidence that BPI promotes retinal pigment epithelial cell growth in the human eye, leading to speculation that it may be of value for the treatment of macular degeneration and related conditions [7]. The BPI family proteins BPIFA1 and BPIFB1 also exhibit antimicrobial and LPS-neutralizing activity, and there is experimental evidence for their roles in nasopharyngeal carcinoma and other respiratory diseases [8,9,10,11]. These prior studies thus highlight *BPI* as a broadly active antimicrobial peptide with a wide range of potential clinical applications.

Enterotoxigenic *Escherichia coli* (ETEC) is a primary bacterial driver of diarrhea in young animals, with F18 fimbriae-expressing ETEC being the main cause of post-weaning diarrhea in piglets [12]. Owing to their immature immune systems, piglets are unable to mount a robust adaptive immune response to ETEC infections, making it challenging to control this disease [13,14]. This has led to an increasing interest in the study of antimicrobial genes associated with porcine resistance to ETEC infections. Zhu et al. [15] compared *BPI* expression in Sutai piglets as a function of age and found that it was expressed at significantly higher levels in duodenal tissue samples from 35-day-old piglets relative to samples from animals in other analyzed age groups. We have also previously explored the link between *BPI* expression and resistance to *E. coli* F18 infections. Sun et al. [16] determined that the demethylation of CpG islands within the *BPI* gene led to its enhanced expression, thereby improving the ability of intestinal tissues to resist *E. coli* F18 infection. Liu et al. [17] further determined that a polymorphism in exon 10 of porcine *BPI* was linked to *E. coli* F18 resistance. To expand upon these prior findings, in this study we sought to confirm the role of *BPI* as a mediator of antimicrobial resistance at a cellular level. As such, we measured *BPI* expression in porcine kidney (PK15) cells following LPS or *E. coli* stimulation. We then generated *BPI*-overexpressing PK15 cells and evaluated the effect of such overexpression on *E. coli* F18 adhesion in vitro through colony counting and qPCR analyses. We have also previously utilized high-throughput data mining and functional validation approaches to identify the molecular mechanisms governing resistance to this ETEC strain, with both the Toll-like receptor signaling and glycosphingolipid biosynthesis-globo series pathways being closely associated with such resistance [18,19]. Herein, we expanded upon these prior results by assessing the impact of *BPI* on the expression of key genes related to the Toll-like receptor 4 (TLR4) signaling pathway (*TLR4, TNF-α*, *IL-1β*, *CD14* and *MyD88*), TLR4 signaling-related proinflammatory cytokines (*IFN-α*, *IFN-β*, *IL-6*, *MIP-1α* and *MIP-1β*) and glycosphingolipid biosynthesis-related genes (*FUT1* and *FUT2*). The overall aim of this study was to clarify the mechanisms whereby *BPI* facilitates porcine *E. coli* F18 resistance, with the goal of thereby supporting future research.

## 2. Materials and Methods

### 2.1. Experimental Materials

PK15 cells were from the American Type Culture Collection (Manassas, VA, USA). Standard porcine *E. coli* F18 strains with the F18ab 107/86 (O139:K12:H1) and F18ac 2134(O157:H19) fimbriae were provided by the veterinary laboratory at the Institute of Microbiology, University of Pennsylvania. LPS was purchased from Sigma-Aldrich (St. Louis, MO, USA). Dulbecco’s Modified Eagle’s Medium (DMEM), Fetal bovine serum (FBS) and trypsin-EDTA were from Gibco (Grand Island, NY, USA). Puromycin and TRIzol were purchased from Thermo Fisher Scientific (Waltham, MA, USA). Phosphate-buffered saline (PBS) was from Beijing Labgic Technology Co., Ltd. (Beijing, China). An enzyme-linked immunosorbent assay (ELISA) kit (Porcine Interferon-α Assay Kit, Porcine Interferon-β Assay Kit, Porcine Interleukin-6 Assay Kit, Porcine Macrophage Inflammatory Protein-1α Assay Kit, Porcine Macrophage Inflammatory Protein-1β Assay Kit) was obtained from Nanjing Jiancheng Bioengineering Institute (Nanjing, China). A DNA extraction kit (DP304) was from Tiangen Biotech Co., Ltd. (Beijing, China), while reverse transcription and real-time fluorescence quantitative kits (AceQ Universal SYBR qPCR Master Mix) were obtained from Vazyme Biotech Co., Ltd. (Nanjing, China).

### 2.2. BPI Overexpression in PK15 Cells 

A *BPI* overexpression lentiviral vector (pGLV5-BPI) and a corresponding negative control (pGLV5-NC) were prepared by GenePharma (Suzhou, China). Prior to lentiviral transduction, PK15 cells were plated at 5.0 × 10^5^ cells/well in 12-well plates in DMEM containing 10% FBS and were grown at 37 °C in a 5% CO_2_ incubator until 80% confluent, at which time four replicate samples were each transduced with the pGLV5-BPI or pGLV5-NC lentiviral vectors. Cells were incubated for 48 h following transduction, at which time positive cells were identified via fluorescence microscopy. Puromycin (10 µg/mL every 24 h) was then used to select for pGLV5-BPI-positive cells, and qPCR was used to confirm successful *BPI* overexpression in these cells.

### 2.3. LPS and E. coli F18 Stimulation

Cells of the blank, pGLV5-NC-positive and pGLV5-BPI-positive cells were plated in 12-well plates (5.0 × 10^5^ per well) until 80% confluent, they were induced with 0.1 µg/mL LPS for 0, 2, 4, 6, 8, 12, 24 and 36 h, and then three replicates were performed per group. Total cellular RNA was extracted to detect Changes in *BPI* gene expression, and cell culture supernatants were collected for ELISA analysis.

Standard porcine *E. coli* strains carrying F18ab and F18ac fimbriae were inoculated into Luria-Bertani (LB) medium for 12 h in a 37 °C with constant agitation Bacteria were then collected via centrifugation at 3000 rpm for 10 min, washed three times with PBS and diluted to 1.0 × 10^9^ colony-forming units (CFU/mL) in cell culture medium.

### 2.4. Analysis of E. coli F18 Adhesion to PK15 Cells

#### 2.4.1. Colony Counting

Cells of the pGLV5-BPI and pGLV5-NC groups were inoculated to the wells of 12-well cell culture plates at a density of 5.0 × 10^5^ cells/well and cultured until the cells reached approximately 80%. Diluents were made from the precipitation residues of the two *E. coli* strains, and 1 mL of the diluent was added to each well, with three replicates per group. The plates were incubated in a 5% CO_2_ incubator at 37 °C for 2 h. After discarding the culture medium, the cells were washed thrice with PBS and immediately treated for 20 min with 0.5% Triton X-100 (prepared with ultrapure water). After serially diluting the culture ten times, LB agar plates were coated with the culture and incubated overnight at 37 °C. Finally, bacterial count was determined by counting the number of colonies on the plate coated with 1000× the bacterial diluent using ImageJ software. The final number of bacteria adhering to the plate (CFU/mL) was equal to the number of colonies on the plate × 10^3^.

#### 2.4.2. Fluorescence Quantitative Polymerase Chain Reaction

Cells of the pGLV5-NC-positive and pGLV5-BPI-positive groups were seeded at 5.0 × 10^5^ cells/well in 12-well plates until 80% confluent, at which time 1 mL of either of the two experimental *E. coli* strains was added to each well. Cells were then incubated for 1 h at 37 °C, after which supernatants were discarded, and cells were washed three times with PBS. Total DNA was then isolated using a DNA extraction kit. After extraction, this DNA was used as the amplification template, and qPCR primers were designed based on the *PILIN* gene of *E. coli* F18ab and F18ac, and the porcine *β-ACTIN* gene, which were detected by fluorescence quantitative polymerase chain reaction (PCR). [20]. All analyses were conducted in triplicate.

### 2.5. Primer Design

qPCR primers for *BPI*, *TLR4*, *MyD88*, *CD14*, *TNF-α*, *IL-1β*, *FUT1*, *FUT2* and *PILIN* were designed with Primer Premier 5.0 based upon sequences in GenBank. *GAPDH* and *β-ACTIN* served as reference controls. All primer synthesis was conducted by Sangon Biotechnology (Shanghai, China), and the corresponding sequences are shown in Table 1.

### 2.6. RNA Extraction and Preparation

TRIzol was used to extract total RNA based upon provided protocols, after which formaldehyde denaturing gel electrophoresis was conducted to gauge RNA integrity, and a NanoDrop ND-1000 spectrophotometer (Thermo Fisher Scientific, Waltham, MA, USA) was employed to assess RNA concentration and purity.

Isolated RNA was then reverse transcribed to produce cDNA in reactions containing 2 µL of 5× qRT SuperMix II, 500 ng of total RNA and up to 10 µL of RNase-free H_2_O. Thermocycler settings were as follows: 25 °C for 10 min, 50 °C for 30 min and 85 °C for 5 min. After preparation, cDNA was stored at 4 °C.

### 2.7. qPCR

All qPCR reactions were conducted in a 20-µL volume composed of 2 µL of cDNA, 0.4 µL of each primer (10 µmol/L), 10 µL of 2 × AceQ Universal SYBR qPCR Master Mix and 7.2 µL of ddH_2_O. Thermocycler settings were as follows: 95 °C for 5 min; 40 cycles of 95 °C for 5 s, 60 °C for 30 s. Melting curves were then used to confirm amplified product specificity. Three independent experimental replicates were conducted for all analyses.

### 2.8. Cytokine ELISAs

We obtained culture supernatants of pGLV5-NC-positive and pGLV5-BPI-positive groups at appropriate time points after LPS stimulation and measured the levels of proinflammatory cytokines (IFN-α, IFN-β, IL-6, MIP-1α and MIP-1β) via ELISA based on the provided kit instructions.

### 2.9. Statistical Analysis

The 2^–ΔΔCt^ method was used to quantify relative gene expression [21], which was normalized to appropriate internal control genes. SPSS 25.0 (SPSS, Inc., Chicago, IL, USA) was used to compare data via ANOVAs with LSD tests. * *p* < 0.05, ** *p* < 0.01 and *** *p* < 0.001.

## 3. Results

### 3.1. LPS and E. coli F18 Induce BPI Upregulation in PK15 Cells

We began by treating PK15 cells with 0.1 µg/mL LPS for 0, 2, 4, 6, 8, 12, 24 and 36 h in order to evaluate the impact of such treatment of *BPI* expression, revealing that this gene was rapidly upregulated following stimulation for 4 h (Figure 1a). When cells were instead stimulated with *E. coli* F18ab or F18ac (1.0 × 10^9^ CFU/mL), we found that the F18ac strain markedly enhanced *BPI* expression (*p* < 0.001), whereas F18ab strain stimulation had no impact on the expression of this gene (*p* > 0.05) (Figure 1b).

### 3.2. Preparation of BPI-Overexpressing PK15 Cells

Next, we confirmed that we were able to successfully transduce PK15 cells with the pGLV5-BPI and pGLV5-NC lentiviral vectors, as confirmed based on the presence of detectable green fluorescent protein within these cells (Figure 2a). Subsequent qPCR analyses confirmed that cells transduced with the pGLV5-BPI plasmid exhibited significant *BPI* overexpression (5556-fold higher than in control cells; Figure 2b). These findings thus indicated that we had successfully prepared *BPI*-overexpressing PK15 cells, which were then used for a series of experiments.

### 3.3. The Impact of BPI Overexpression on E. coli F18 Adhesion to PK15 Cells

A colony counting assay revealed that the overexpression of *BPI* was sufficient to markedly suppress the adhesion of *E. coli* F18ab (Figure 3a) and *E. coli* F18ac (Figure 3b) to PK15 cells (*p* < 0.01 and *p* < 0.001, respectively). In line with this, qPCR analyses of *PILIN* gene confirmed that *BPI* overexpression significantly impaired *E. coli* F18ab (Figure 3c) and *E. coli* F18ac (Figure 3d) adhesion to these PK15 cells (*p* < 0.01 and *p* < 0.001, respectively). As such, these findings indicate that *BPI* overexpression can interfere with *E. coli* F18 adherence to PK15 cells.

### 3.4. The Impact of BPI Overexpression on TLR4 and Glycosphingolipid Biosynthesis-Globo Series Pathway-Related Gene Expression

Next, we assessed the expression of the glycosphingolipid biosynthesis-globo series pathway genes *FUT1* and *FUT2* in control or *BPI*-overexpressing PK15 cells, in which we also evaluated the expression of TLR4 signaling pathway-related genes (*TLR4, TNF-α*, *IL-1β*, *CD14* and *MyD88*). We found that *BPI* overexpression did not impact *TLR4*, *CD14*, *TNF-α*, *IL-1β*, *MyD88*, *FUT1* or *FUT2* expression (*p* > 0.05). We also found that *MyD88*, *FUT1* and *FUT2* were highly expressed in PK15 cells (Figure 4).

### 3.5. The Impact of BPI Overexpression on the Upregulation of TLR4-Related Proinflammatory Cytokines

Lastly, we evaluated the production of *BPI*-overexpressing or control PK15 cells in response to LPS treatment (0.1 µg/mL for 0–36 h). ELISAs revealed that supernatant levels of proinflammatory cytokines such as *IFN-α*, *IFN-β*, *IL-6*, *MIP-1α* and *MIP-1β* initially rose and then declines over time. Furthermore, we found that *BPI* overexpression significantly reduced the LPS-induced upregulation of the TLR4-related proinflammatory cytokines *IFN-α*, *IFN-β*, *IL-6*, *MIP-1α* and *MIP-1β* in these cells (Figure 5).

## 4. Discussion

*E. coli* F18 can infect porcine cells and is characterized in part by the presence of high levels of cell wall-associated LPS [22]. Prior genetic analyses have demonstrated that the ability of piglets to resist *E. coli* F18 infection is tied to both innate immunity and the expression of the *E. coli* F18 receptor by intestinal epithelial cells [23,24,25]. PK15 cells have commonly been utilized as a model cell line for analyses of pathogenic *E. coli* adhesion and associated immune responses [26,27]. As such, we stimulated PK15 cells with LPS or with porcine *E. coli* F18, revealing that both of these treatments resulted in significant *BPI* upregulation. Glycosphingolipid biosynthesis-globo series pathway genes (*FUT1*, *FUT2*) are involved in the formation of the *E. coli* F18 receptor, and its expression level is closely related to resistance to *E. coli* F18 in piglets [28,29]. TLRs recognize different microbial components, sense microbial populations in the intestinal tract, initiate proinflammatory signaling pathways to resist the invasion of pathogenic microorganisms and play an important role in immune regulation in the process of resisting *E. coli* F18 infection [30,31]. To further assess the mechanistic role of *BPI* in the context of *E. coli* F18 infection, we next generated *BPI*-overexpressing PK15 cells. While such overexpression did not alter the mRNA level expression of *FUT1*, *FUT2* or TLR4 signaling pathway-related genes (*TLR4*, *CD14*, *TNF-*α, *IL-1β*, *IFN-α* and *MyD88*), it did markedly impair *E. coli* F18 adhesion to these cells in vitro. We also found that *BPI* overexpression suppressed LPS-induced *IFN-γ*, *IFN-α*, *IFN-β*, *IL-6, MIP-1α* and *MIP-1β* expression. Balakrishnan et al. [32] have previously shown that LPS can increase *BPI* protein levels within cells. At baseline, proinflammatory cytokine production is minimal, whereas it is rapidly induced at high levels in response to LPS stimulation. *BPI* can bind to the conserved LPS lipid A/inner core, and it can thereby inhibit its ability to interact with TLR4 and to thereby activate proinflammatory signaling [33,34]. We therefore speculate that *BPI* inhibits the LPS-mediated induction of the host cellular immune response by forming a complex with LPS, thereby enabling cells to resist *E. coli* F18 infection.

We confirmed the link between *BPI* expression and *E. coli* F18 infection at the cellular level in porcine PK15 cells in vitro, underscoring the role of *BPI* as an inhibitor of inflammatory responses at least in part owing to its ability to neutralize LPS, thereby enabling these cells to better resist infection by this ETEC strain. Our data further demonstrate that *BPI* can markedly reduce *E. coli* adherence to PK15 cells in vitro. Future GST pull-down, co-immunoprecipitation (Co-IP) and gene knockout experiments will be necessary in order to fully understand the mechanisms whereby *BPI* influences *E. coli* F18 receptor molecules in this experimental context. It is also important to note that IFN-α/β are key antiviral cytokines that can induce an antiviral state in both infected and uninfected adjacent cells [35,36] by upregulating a range of IFN-stimulated genes with diverse antiviral activities [37,38]. As we found that *BPI* regulates IFN-α/β production by PK15 cells, this suggests that it may additionally serve as a potential regulator of antiviral immunity in these porcine cells. However, future experimental work will be needed to test this possibility.

## 5. Conclusions

In conclusion, we found that the overexpression of porcine *BPI* significantly reduced the adhesion of *E. coli* F18 to porcine kidney cells in vitro, although it had no impact on the expression of TLR4 or glycosphingolipid biological signaling pathway-related genes in these cells. In addition, *BPI* overexpression was sufficient to markedly suppress the LPS-induced upregulation of TLR4 signaling-related proinflammatory cytokines including *IFN-α*, *IFN-β*, *MIP-1α*, *MIP-1β* and *IL-6*.

## Figures and Tables

**Figure 1 animals-10-02118-f001:**
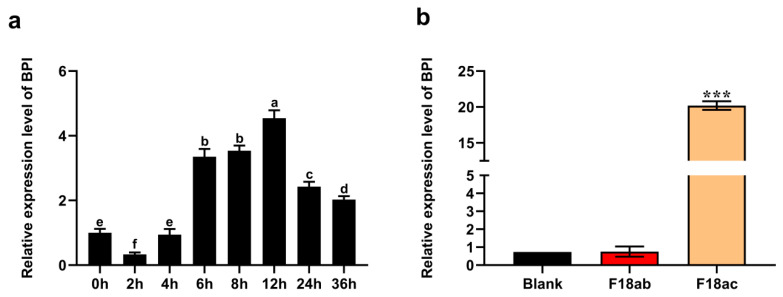
The impact of LPS and *E. coli* F18 stimulation on *BPI* gene expression. (**a**) LPS was used to stimulate porcine kidney (PK15) cells for 0, 2, 4, 6, 8, 12, 24 and 36 h, after which bactericidal/permeability-increasing protein (*BPI*) expression was analyzed. Different lowercase letters indicate significant differences (*p* < 0.05), whereas identical letters indicate a lack of any difference between groups (*p* > 0.05). (**b**) *BPI* expression was assessed in PK15 cells following stimulation with *E. coli* F18ab or F18ac. Blank: untreated control group. Samples were analyzed in triplicate. Data are means ± standard deviation. *** *p* < 0.001.

**Figure 2 animals-10-02118-f002:**
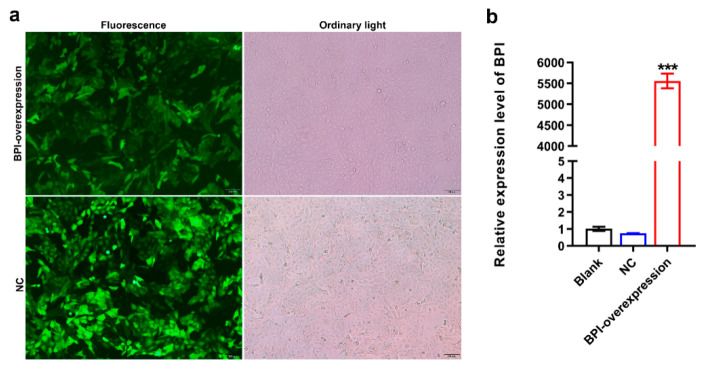
Preparation of *BPI*-overexpressing PK15 cells. (**a**) Green fluorescent protein (GFP) expression in the indicated PK15 cells was assessed at 96 h following lentiviral transduction. (**b**) *BPI* expression was measured via qPCR. Blank: untreated control group, NC: PK15 cells transfected with the negative control lentivirus (pGLV5-NC), BPI-overexpression: PK15 cells transfected with the *BPI* gene overexpression lentivirus (pGLV5-BPI). Samples were analyzed in triplicate. Data are means ± standard deviation. *** *p* < 0.001.

**Figure 3 animals-10-02118-f003:**
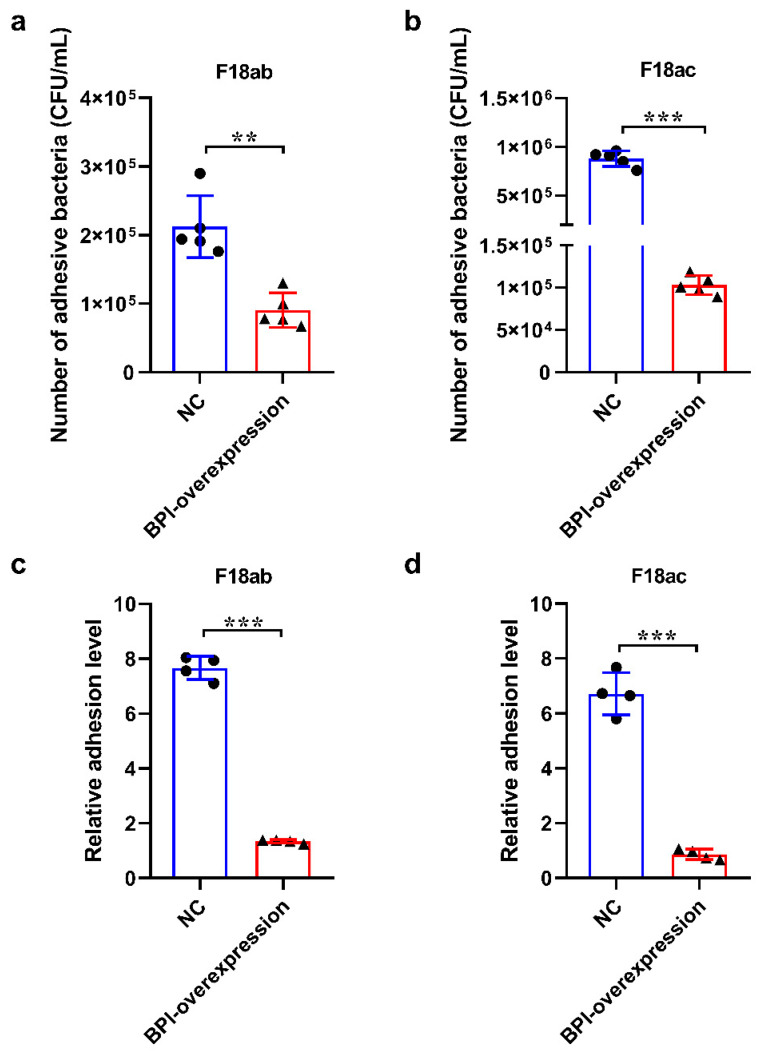
The impact *BPI* overexpression on *E. coli* F18 adhesion to PK15 cells in vitro. (**a**,**b**) *E. coli* F18ab and F18ac adhesion to PK15 cells was assessed via a colony counting approach (*n* = 5). (**c**,**d**) *E. coli* F18ab and F18ac adhesion to PK15 cells was assessed via qPCR. NC: PK15 cells transfected with the negative control lentivirus (pGLV5-NC), BPI-overexpression: PK15 cells transfected with the *BPI* gene overexpression lentivirus (pGLV5-BPI). Data are means ± standard deviation. (*n* = 4). ** *p* < 0.01, *** *p* < 0.001.

**Figure 4 animals-10-02118-f004:**
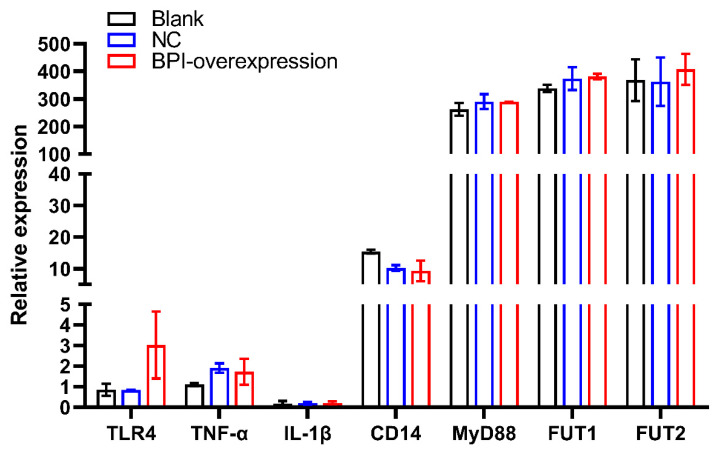
The impact of *BPI* overexpression on Toll-like receptor 4 (TLR4) signaling pathway-related gene expression and on α-(1,2) fucosyltransferase (*FUT1*, *FUT2*) gene expression. Blank: untreated control group, NC: PK15 cells transfected with the negative control lentivirus (pGLV5-NC), BPI-overexpression: PK15 cells transfected with the *BPI* gene overexpression lentivirus (pGLV5-BPI). Samples were analyzed in triplicate. Data are means ± standard deviation.

**Figure 5 animals-10-02118-f005:**
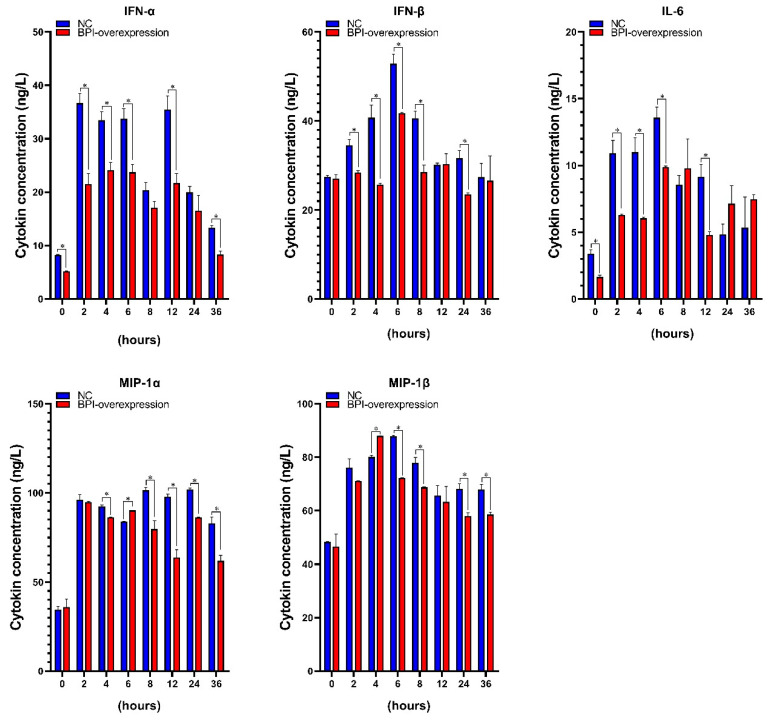
The impact of LPS stimulation on TLR4-related inflammatory cytokine production. NC: PK15 cells transfected with the negative control lentivirus (pGLV5-NC), BPI-overexpression: PK15 cells transfected with the *BPI* gene overexpression lentivirus (pGLV5-BPI). Samples were analyzed in triplicate. Data are means ± standard deviation. * *p* < 0.05.

**Table 1 animals-10-02118-t001:** qPCR primer sequences.

Gene	Primer Sequence	Product Length (bp)
*BPI* *TLR4*	F: 5′-ATATCGAATCTGCGCTCCGA-3′ R: 5′-TTGATGCCAACCATTCTGTCC-3′ F: 5′-CAGATAAGCGAGGCCGTCATT-3′ R: 5′-TTGCAGCCCACAAAAAGCA-3′	136 113
*MyD88*	F: 5′-GTGCCGTCGGATGGTAGT-3′ R: 5′-CAGTGATGAACCGCAGGAT-3′	173
*CD14*	F: 5′-CCTCAGACTCCGTAATGTG-3′ R: 5′-CCGGGATTGTCAGATAGG-3′	180
*TNF-α*	F: 5′-CGACTCAGTGCCGAGATCAA-3′ R: 5′-CCTGCCCAGATTCAGCAAAG-3′	58
*IL-1β* *FUT1* *FUT2* *GAPDH* *PILIN*	F: 5′-TGATTGTGGCAAAGGAGGA-3′ R: 5′-TTGGGTCATCATCACAGACG-3′ F: 5′-TTTTAAGCCCCCAAACTGCC-3′ R: 5′-TAAATCGACCCCATCAGCCTC-3′ F: 5′-AATCCCTGACCTCACTCCGTG-3′ R: 5′-CGGAACTACAACTGCTGGCC-3′ F: 5′-ACATCATCCCTGCTTCTACTGG-3′ R: 5′-CTCGGACGCCTGCTTCAC-3′ F: 5′-AGGCCGAACCAAAGAAGCAT-3′ R: 5′-TCACCATCAGGGTTTCTGAGT-3′	63 126 123 188 117
*β-ACTIN*	F: 5′-GTCGTACTCCTGCTTGCTGAT-3′ R: 5′-CCTTCTCCTTCCAGATCATCGC-3′	119

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
