# Peer review of "The Impact of BPI Expression on Escherichia coli F18 Infection in Porcine Kidney Cells"

_animals, 2020, doi:10.3390/ani10112118_

Round 1
Reviewer 1 Report
After considering the reviewers suggestions, the authors submitted an improved version of the manuscript.
Author Response
Dear reviewer:
Thank you again for your comments. Your comments make our paper more perfect and also have an important facilitating effect on our research.
Sincerely yours,
Wenbin Bao
Key Laboratory for Animal Genetics, Breeding, Reproduction and Molecular Design of Jiangsu Province, College of Animal Science and Technology, Yangzhou University, Yangzhou, 225009 Jiangsu, China.
Phone: 86-514-87979316
Fax: 86-514-87350440
E- mail: [email protected]

Reviewer 2 Report
THe manuscript was improved in many parts. Several issues have been addressed and solved, along with the English language, which is now fine, with just some minor details to be further improved. All the major corrections were adopted, and authors were also exhaustive in providing many additional information on some points.
The paper is nearly ready for publication, even if some minor corrections, especially regarding materials and methods, are still required.
Abstract:
- Line 27: remove the “to” between “E. coli F18” and “adherence”
- Lines 28-32: authors should rephrase this part because it seems “wordy”. Maybe two distinct sentences could work better.
Introduction:
- Line 41: the word “via” should be corrected with the word “by”.
- Line 54: the word “man” should be corrected to “main”
Materials and methods:
- Paragraph 2.1: Authors should include the commercial names of ELISA kits and DNA extraction kits
- Paragraph 2.1: Authors should add some additional information about the two coli F18 strains they employed in this study. Are they field strains? How were they obtained?
- Paragraph 2.3: This paragraph still seems a little bit generic and is not well clear to which parts of the results it refers to. Authors should explain better what “analysis” they refer to at the end of line 101 and what was the purpose of the procedure of lines 102-105 because it is not clear the link with the results part. Moreover, authors should explain how they assessed bacterial concentration in order to dilute the bacterial culture up to a concentration of 1x10^9 CFU/mL.
- Paragraph 2.4: In our comments, authors stated that they added a sentence regarding proinflammatory cytokines assessment that we could not find in the text. They should add it to the text.
- Paragraph 2.4.1: It is not clear how the two coli strains were added to the cultured cells (lines 110-111). Authors should be more precise in explaining this part of the experimental procedure.
- Paragraph 2.4.2: We really appreciate the explanation authors provided us to understand their adoption of the qPCR method to assess the bacterial adhesion to PK15 cells. However, we strongly suggest to modify the reference to “gene expression” in this part (and in the results as well) because “gene expression” is usually linked to an analysis base on RNA instead of DNA. Moreover, we suggest to rephrase lines 121-122 because they are not clear.
- Paragraph 2.7: In our comments, authors stated that they added a sentence regarding the melting curve that we could not find in the text. They should add it to the text.
- Paragraph 2.8: We suggest to add more information because this paragraph is not well linked to the others. For example, it is not clear to which cell culture supernatants authors refer to.
- Table 1: In our comments, authors explained better the purpose of each gene of the table. They should add these information (housekeeping gene, bacterial gene, cell gene) also in the table.
Results:
- Paragraph 3.1: We believe that “BPI cells” should be corrected to “PK15 cells”.
- Figure 5: We really appreciate the addition of the statistics to figure 5, which now lets the readers to understand the difference between the two experimental groups. However, we suggest to restore the previous graphs (the line charts) by adding the statistical significance by using – for instance – “a, b” letters for each timepoint. This should also improve the “readability” of the graphs.
Discussion:
- Lines 227-232: authors should rephrase this long sentence – for instance by splitting it into shorter sentences – in order to let these results and this part of the discussion to be clearer for all the readers.
Author Response
Response to reviewer’s comments:
Thanks for your comments. Those comments are all valuable and very helpful for revising and improving our paper, as well as the important guiding significance to our researches.
Abstract:
Line 27: remove the “to” between “E. coli F18” and “adherence”
Response: Thanks for your correction. It has been removed. (lines 27 on page 1).
Lines 28-32: authors should rephrase this part because it seems “wordy”. Maybe two distinct sentences could work better.
Response: Thanks for your suggestion. We have revised the sentence as “BPI overexpression had no significant effect on the mRNA-level expression of genes associated with glycosphingolipid biosynthesis or TLR4 signaling. BPI overexpression suppressed the LPS-induced TLR4 signaling pathway-related expression of proinflammatory cytokines (IFN-α, IFN-β, MIP-1α, MIP-1β, and IL-6).” (lines 28-32 on page 1).
Introduction:
Line 41: the word “via” should be corrected with the word “by”.
Response: Thanks for your correction. It has been corrected. (lines 41 on page 2).
Line 54: the word “man” should be corrected to “main”
Response: Thanks for your correction. It has been corrected. (lines 54 on page 2).
Materials and methods:
Paragraph 2.1: Authors should include the commercial names of ELISA kits and DNA extraction kits
Response: Thanks for your advice. We have revised the sentence as “An enzyme-linked immunosorbent assay (ELISA) kit (Porcine Interferon-α Assay Kit, Porcine Interferon-β Assay Kit, Porcine Interleukin-6 Assay Kit, Porcine Macrophage Inflammatory Protein-1α Assay Kit, Porcine Macrophage Inflammatory Protein-1β Assay Kit) was obtained from Nanjing Jiancheng Bioengineering Institute (Nanjing, China). A DNA extraction kit (DP304) was from Tiangen Biotech Co., Ltd. (Beijing, China).” (lines 86-91 on page 2-3).
Paragraph 2.1: Authors should add some additional information about the two coli F18 strains they employed in this study. Are they field strains? How were they obtained?
Response: Thanks for your advice. We have revised the sentence as “Standard porcine E. coli F18 strains with the F18ab 107/86 (O139:K12:H1) and F18ac 2134(O157:H19) fimbriae were provided by the veterinary laboratory at the Institute of Microbiology, University of Pennsylvania.” (lines 80-82 on page 2).
Paragraph 2.3: This paragraph still seems a little bit generic and is not well clear to which parts of the results it refers to. Authors should explain better what “analysis” they refer to at the end of line 101 and what was the purpose of the procedure of lines 102-105 because it is not clear the link with the results part. Moreover, authors should explain how they assessed bacterial concentration in order to dilute the bacterial culture up to a concentration of 1x10^9 CFU/mL.
Response: Thanks for your advice. We have revised the sentence as “Cells of the blank, pGLV5-BPI-positive, and pGLV5-NC-positive cells were plated in 12-well plates (5.0 × 105 per well) until 80% confluent, they were induced with 0.1 µg/mL LPS for 0, 2, 4, 6, 8, 12, 24, and 36 h, three replicates were performed per group. Total cellular RNA was extracted to detect Changes in BPI gene expression. And cells culture supernatants were collected for ELISA analysis.” (lines 104-111 on page3). The microbial concentration in the bacterial solution is directly proportional to the optical density (OD value) of the liquid. Therefore, the optical density of the bacterial suspension is measured by spectrophotometer to infer the concentration of the bacterial solution. This method is one of the conventional methods for determining the concentration of the bacterial solution [1,2].
- Liu, Y.; Jia, Y.; Yang, K.; Tong, Z.; Shi, J.; Li, R.; Xiao, X.; Ren, W.; Hardeland, R.; Reiter, R.J.; et al. Melatonin overcomes MCR-mediated colistin resistance in Gram-negative pathogens. Theranostics 2020, 29, 10697-10711.
- Ma, X.L.; Chen, R.J.; Wang, F.; Xu, T.Y. Determination of Bacterial Suspension Concentration the Scope of Application Using Absorbance Method. Journal of Microbiology 2014, 34, 90-92.
Paragraph 2.4: In our comments, authors stated that they added a sentence regarding proinflammatory cytokines assessment that we could not find in the text. They should add it to the text.
Response: Thanks for your advice. We have added the sentence about proinflammatory cytokines to Paragraph 2.8. “We obtained culture supernatants of pGLV5-NC and pGLV5-BPI groups at appropriate time points after LPS stimulation and measured the levels of proinflammatory cytokines (IFN-α, IFN-β, IL-6, MIP-1α, and MIP-1β) via ELISA based on the provided kit instructions.” (lines 169-173 on page 5).
Paragraph 2.4.1: It is not clear how the two coli strains were added to the cultured cells (lines 110-111). Authors should be more precise in explaining this part of the experimental procedure.
Response: Thanks for your advice. We have revised the sentence as “Cells of the pGLV5-BPI and pGLV5-NC groups were inoculated to the wells of 12-well cell culture plates at a density of 5.0 × 105 cells/well, and cultured until the cells reached approximately 80%. Diluents were made from the precipitation residues of the two E. coli strains, and 1 mL of the diluent was added to each well, with three replicates per group. The plates were incubated in a 5% CO2 incubator at 37 ℃ for 2 h. After discarding the culture medium, the cells were washed thrice with PBS, immediately treated for 20 min with 0.5% Triton X-100 (prepared with ultrapure water). After serially diluting the culture ten times, LB agar plates were coated with the culture and incubated overnight at 37 ℃. Finally, bacterial count was determined by counting the number of colonies on the plate coated with 1000× the bacterial diluent using ImageJ software. The final number of bacteria adhering to the plate (CFU/mL) was equal to the number of colonies on the plate × 103.” (lines 117-127 on page3).
Paragraph 2.4.2: We really appreciate the explanation authors provided us to understand their adoption of the qPCR method to assess the bacterial adhesion to PK15 cells. However, we strongly suggest to modify the reference to “gene expression” in this part (and in the results as well) because “gene expression” is usually linked to an analysis base on RNA instead of DNA. Moreover, we suggest to rephrase lines 121-122 because they are not clear.
Response: Thanks for your advice. We have modified the reference to "gene expression" in this part (and in the results). (lines 136 on page4). We have revised the sentence as “After extraction, this DNA was used as the amplification template, and qPCR primers were designed based on the PILIN gene of E. coli F18ab and F18ac, and the porcine β-ACTIN gene, which were detected by fluorescence quantitative polymerase chain reaction (PCR). (lines 141-145 on page4).
Paragraph 2.7: In our comments, authors stated that they added a sentence regarding the melting curve that we could not find in the text. They should add it to the text.
Response: Thanks for your advice. Melting curves were then used to confirm amplified product specificity. (lines 166-168 on page 5).
Paragraph 2.8: We suggest to add more information because this paragraph is not well linked to the others. For example, it is not clear to which cell culture supernatants authors refer to.
Response: Thanks for your advice. We have revised the sentence as “We obtained culture supernatants of pGLV5-NC and pGLV5-BPI groups at appropriate time points after LPS stimulation and measured the levels of proinflammatory cytokines (IFN-α, IFN-β, IL-6, MIP-1α, and MIP-1β) via ELISA based on the provided kit instructions.” (lines 169-173 on page 5).
Table 1: In our comments, authors explained better the purpose of each gene of the table. They should add these information (housekeeping gene, bacterial gene, cell gene) also in the table.
Response: Thanks for your advice. We have added (housekeeping gene, bacterial gene, cell gene) to the table. BPI, TLR4, MyD88, CD14, TNF-α, IL-1β, FUT1, and FUT2 genes are cellular genes, and GAPDH is the housekeeping gene for cellular genes. PILIN is a bacterial gene and β-ACTIN acts as a housekeeping gene.
Results:
Paragraph 3.1: We believe that “BPI cells” should be corrected to “PK15 cells”.
Response: Thanks for your correction. It has been corrected. (lines 182 on page 5).
Figure 5: We really appreciate the addition of the statistics to figure 5, which now lets the readers to understand the difference between the two experimental groups. However, we suggest to restore the previous graphs (the line charts) by adding the statistical significance by using – for instance – “a, b” letters for each timepoint. This should also improve the “readability” of the graphs.
Response: Thanks for your advice. We used "a, b" letters to increase the statistical significance at each time point in the line charts, but found that "a, b" letters were difficult to significantly mark between the NC group and the BPI-overexpression group, and resulted in poor aesthetic appearance of the line charts. Therefore, for this modification, we believed that the use of bar chart could more intuitively show the significant difference between the NC group and the BPI-overexpression group. If the editor teacher believed that it was necessary to add it, please inform us in time, and we could make modification at any time.
Discussion:
Lines 227-232: authors should rephrase this long sentence – for instance by splitting it into shorter sentences – in order to let these results and this part of the discussion to be clearer for all the readers.
Response: Thanks for your advice. We have revised the sentence as “As such, we stimulated PK15 cells with LPS or with porcine E. coli F18, revealing that both of these treatments resulted in significant BPI upregulation. Glycosphingolipid biosynthesis-globo series pathway genes (FUT1, FUT2) are involved in the formation of the E. coli F18 receptor, and its expression level is closely related to resistance to E. coli F18 in piglets [28,29]. TLRs recognize different microbial components, sense microbial populations in the intestinal tract, initiate proinflammatory signaling pathways to resist the invasion of pathogenic microorganisms, and play an important role in immune regulation in the process of resisting E. coli F18 infection”. (lines 255-264 on page 12-13).

This manuscript is a resubmission of an earlier submission. The following is a list of the peer review reports and author responses from that submission.
Round 1
Reviewer 1 Report
The authors performed one very interesting study evaluating the effect of BPI expression on Escherichia coli F18 infection in porcine kidney cells. They used two different approach stimulating the expression of BPI with LPS, and overexpression of BPI with transfected plasmids. Interesting they found different results when used these two methodologies. In their conclusion they highlighted major findings:
- overexpression the porcine BPI gene can significantly reduce the adhesion level of porcine kidney cells to coli F18 in vitro, but had no significant effect on the expression of key genes of the TLR4 signaling pathway and the glycosphingolipid biological signaling pathway,
- LPS induced overexpression of the BPI gene could significantly reduce the levels of proinflammatory cytokines (IFN-α, IFN-β, MIP-1α, MIP-1β, and IL-6) in the TLR4 signaling pathway.
In my opinion the results only corroborate part of the first statement. “Overexpression the porcine bpi gene can significantly reduce the adhesion level of E.coli F18 to porcine kidney cells in vitro.”
My suggestion is a detailed review of the results showed in figures 4 and 5 because mainly in figure 5 the error bars indicated some problem in the experiments.
Author Response
Response to reviewer’s comments:
My suggestion is a detailed review of the results showed in figures 4 and 5 because mainly in figure 5 the error bars indicated some problem in the experiments.
Response: Thanks for your comments. Those comments are all valuable and very helpful for revising and improving our paper, as well as the important guiding significance to our researches. We have reviewed the results of figure 4 with figure 5, and modified the line chart of figure 5 to a bar chart to more intuitively show the effects of overexpression of BPI gene on cytokine concentrations.

Reviewer 2 Report
The manuscript “Insight into the effect of BPI expression on Escherichia coli F18 infection in porcine kidney cells” deals with the investigation of the relationship between the expression of the antibacterial protein BPI and the adhesion of E. coli F18 to cells, also providing insights on the expression of inflammatory cytokines or cellular pathways demonstrated to be potentially related to E. coli F18 infection.
The overall result appears to be interesting and capable to complement data obtained by other studies, however some points should be further clarified by authors. Moreover, some additional explanations or tests are requested in order to better justify some statements reported throughout discussions and conclusions, thus requiring some major revisions.
Introduction
An adequate introduction to ETEC strains and Escherichia coli F18 should be added in this section to better contextualize the issue related to this pathogen. Very little information is provided throughout the introduction, but a small paragraph should be added to let the readers to understand why E. coli F18 is challenging for pig breeding.
Lines 54 – The expression “is pretty hot” seems inappropriate and should be changed.
Lines 56-58 – Reference should be provided.
Line 59 – “Improve” is repeated two times, a synonym should be chosen.
Line 70 – Please remove the capital letter in “glycosphingolipid”
Why have authors decided to use a porcine kidney cell line (PK15) to perform this study, instead of opting for another cellular line like IPEC-J2 or IPI cells, which are very established porcine intestinal cell lines and therefore probably more suitable to investigate the mechanism of action of an intestinal pathogen?
Materials and Methods
In general, the order in which materials and methods are presented should follow the way results are presented. Authors should therefore reorder this section.
Line 86-88 – More details about the precise ELISA kits and DNA extraction kit used to perform these tests should be provided.
Line 99 – Authors should explain how they assessed the effective overexpression of BPI in transfected PK15 cells: in the results, a graph regarding a gene expression test is shown, but no explanation on how it was made is provided in paragraph 2.2.
Paragraph 2.3 – Two different tests (PK15 stimulation with LPS and PK15 infection with F18) are presented in this paragraph, but in the materials and methods explanation they seem to be the same test. Authors should rewrite this part to better clarify the two different protocols used.
Line 105 – Usually, in experiments concerning infection of cells with microorganisms, an active bacterial culture is preferred. Why did authors decide to infect cells with a 12h bacterial culture (potentially, a stationary or decline phase of the culture), instead of an active culture in the mid or late log phase?
Paragraph 2.4 – It is not clear which cells (“blank” cells or transfected cells) were treated with LPS and monitored for cytokine production, because it is only reported in results. Authors should add this indication also in this paragraph. Moreover, authors should clarify why they did not include the “blank” group in this experiment.
Lines 117-119 – It is not clear what is the “diluent” in this sentence. Authors should clarify this point by rewriting it.
Lines 122-125 – The counting of the bacterial colonies is not clear. Moreover, it is not clear why authors expressed the result by plotting the number of colonies x 103 instead of a clearer log10 CFU/mL.
Line 130 – Why did authors perform this test by incubating cells with E. coli F18 for only 1 hour instead of keeping the incubation time at 2 hours as for the colony count test?
Line 132-135 – It is not clear why authors perform a gene expression assay on a DNA sample. It seems more likely a relative quantification of bacteria by evaluating the amount of DNA. Authors should better clarify this part, in particular the result that they show related to the PILIN gene evaluation.
Paragraph 2.6 – Authors should clarify which are the bacterial and cellular genes, with their respective housekeeping genes. Moreover, did authors check if the PCR amplification products matched the expected amplicon?
Line 143 – Have authors included a purification step for RNA in order to remove all the DNA in the sample?
Line 148 – Why authors stored cDNA at +4°C, while it is usually stored at -20°C?
Lines 153-154 – The protocol of the melting curve is not clear. Authors should revise it.
Paragraph 2.9 – Authors should add little information about the statistical tests employed in the study to determine statistical significance.
Results
Figure 1 – Authors should comment on the reason why data show a lower BPI expression at 2h. Moreover, authors should also comment on the fact that only E. coli F18ac can significantly increase BPI expression, while F18ab does not exert differences in relative expression levels of the gene.
Lines 174-175 – Authors should also add indications about the significance level of the different letters reported in graph 1a. Is there a P < 0.001?
Figure 2 – While the pictures of transfected PK15 cells are taken at 96h, it is not clear when the gene expression analysis of BPI has been performed. Moreover, how long is the expression constant and stable? These are all relevant information to allow a complete understanding on the compatibility of the BPI expression with the timeframes selected for the experiments with transfected PK15 cells.
Figure 5 – In lines 224-226, authors state that the overexpression of the BPI gene can significantly reduce the concentration of several cytokines. In order to confirm this, authors should add a statistical analysis to all the displayed graphs, otherwise the statement cannot be considered true.
Discussion and conclusions
After having addressed all the suggested reviews for the previous parts, authors should revise discussion accordingly, by paying particular attention to the reason why they selected PK15 cells instead of other porcine intestinal cell lines.
Moreover, authors should reconsider the statement at lines 251-252 accordingly to the statistical analysis that they should perform on the data shown in figure 5. All the statement and the considerations from line 252 to 259 can be justified and proved only if the statistical analysis shows significant differences between the experimental groups.
The same applies for the final sentence of the conclusions.
Author Response
Response to reviewer’s comments:
Introduction
An adequate introduction to ETEC strains and Escherichia coli F18 should be added in this section to better contextualize the issue related to this pathogen. Very little information is provided throughout the introduction, but a small paragraph should be added to let the readers to understand why E. coli F18 is challenging for pig breeding.
Response: Thanks for your advice. We have provided more introductions to Escherichia coli F18 in this section (lines 56-61 on page 2).
Lines 54 – The expression “is pretty hot” seems inappropriate and should be changed.
Response: Thanks for your comments. It has been removed.
Lines 56-58 – Reference should be provided.
Response: Thanks for your advice. The reference has been provided (lines 62 on page 2).
Line 59 – “Improve” is repeated two times, a synonym should be chosen.
Response: Thanks for your advice. We have changed “improving” to “increasing” (lines 66 on page 2).
Line 70 – Please remove the capital letter in “glycosphingolipid”
Response: Thanks for your suggestion. We have changed the capital letter in “glycosphingolipid” (lines 78 on page 2).
Why have authors decided to use a porcine kidney cell line (PK15) to perform this study, instead of opting for another cellular line like IPEC-J2 or IPI cells, which are very established porcine intestinal cell lines and therefore probably more suitable to investigate the mechanism of action of an intestinal pathogen?
Response: Thanks for your advice. We provide relevant explanations in the discussion section (lines 253-255 on page 9). The key to Escherichia coli (E. coli) infection lies in the expression of E. coli receptor in the porcine intestinal epithelial cells and the difference of innate immunity. At present, the research on E. coli mainly focus on two categories, the screening of E. coli receptors and immune-related genes. The porcine intestinal epithelial cell line (IPEC-2) is a very suitable in vitro cell model of E. coli receptor genes [1,2]. This study discusses the effect of porcine BPI gene on E. coli F18 infection and the exploration of immune-related functions. The porcine epithelial kidney cell line (PK15) serve as an in vitro cell model for studying immune responses and validating the adhesion pattern of pathogenic E. coli [3,4]. Based on the function of BPI gene and the aim of this study, we selected PK15 cells as the in vitro cell model for BPI gene in this study.
- Cirl, C.; Wieser, A.; Yadav, M.; Duerr, S.; Schubert, S.; Fischer, H.; Stappert, D.; Wantia, N.; Rodriguez, N.; Wagner, H.; et al. Subversion of Toll-like receptor signaling by a unique family of bacterial Toll/interleukin-1 receptor domain-containing proteins. Nat Med 2008, 14, 399-406.
- Wu, Z.; Liu, Y.; Dong, W.; Zhu, G.; Wu, S.; Bao, W. CD14 in the TLRs signaling pathway is associated with the resistance to E. coli F18 in Chinese domestic weaned piglets. Sci Rep 2016, 6, 24611.
- Jiang, S.; Li, F.; Li, X.; Wang, L.; Zhang, L.; Lu, C.; Zheng, L.; Yan, M. Transcriptome analysis of PK-15 cells in innate immune response to porcine deltacoronavirus infection. PLoS One 2019, 14, e0223177.
- Frommel, U.; Bohm, A.; Nitschke, J.; Weinreich, J.; Groß, J.; Rödiger, S.; Wex, T.; Ansorge, H.; Zinke, O.; Schröder, C.; et al. Adhesion patterns of commensal and pathogenic Escherichia coli from humans and wild animals on human and porcine epithelial cell lines. Gut Pathog 2013, 5, 31.
Materials and Methods
In general, the order in which materials and methods are presented should follow the way results are presented. Authors should therefore reorder this section.
Response: Thanks for your advice. The order of this section has been modified.
Line 86-88 – More details about the precise ELISA kits and DNA extraction kit used to perform these tests should be provided.
Response: Thanks for your advice. We performed the experiments in strict accordance with the manufacturer’s instructions of the ELISA kits and DNA extraction kit, which are listed in 2.1 (lines 93-95 on page 2).
Line 99 – Authors should explain how they assessed the effective overexpression of BPI in transfected PK15 cells: in the results, a graph regarding a gene expression test is shown, but no explanation on how it was made is provided in paragraph 2.2.
Response: Thanks for your suggestion. We have illustrated it in paragraph 2.2 that we employed a qPCR approach to examine the overexpression efficiency of the BPI gene in PK15 cells (lines 107-108 on page 3).
Paragraph 2.3 – Two different tests (PK15 stimulation with LPS and PK15 infection with F18) are presented in this paragraph, but in the materials and methods explanation they seem to be the same test. Authors should rewrite this part to better clarify the two different protocols used.
Response: Thanks for your suggestion. We rewrite the content of this paragraph (lines 110-118 on page 3).
Line 105 – Usually, in experiments concerning infection of cells with microorganisms, an active bacterial culture is preferred. Why did authors decide to infect cells with a 12h bacterial culture (potentially, a stationary or decline phase of the culture), instead of an active culture in the mid or late log phase?
Response: The bacteria grew rapidly in the logarithmic phase, the number of living bacteria increased in a constant geometric series, and the logarithm of the number of bacteria on the growth curve increased linearly. In this stage, the morphology, staining and physiological activity of bacteria are typical, and they are sensitive to the action of external environmental factors. Therefore, we choose this stage, the general logarithmic phase of bacteria is 8-18 h after culture. E. coli can double its population every 20 minutes [1], and the number of bacteria produced by 12 h culture can completely meet our experimental requirements.
- Allen, R.J.; Waclaw, B. Bacterial growth: a statistical physicist's guide. Rep Prog Phys 2019, 82, 016601.
Paragraph 2.4 – It is not clear which cells (“blank” cells or transfected cells) were treated with LPS and monitored for cytokine production, because it is only reported in results. Authors should add this indication also in this paragraph. Moreover, authors should clarify why they did not include the “blank” group in this experiment.
Response: Thanks for your advice. We have revised the sentence as “We obtained culture supernatants of pGLV5-NC and pGLV5-BPI groups at various time points after LPS stimulation and measured the levels of proinflammatory cytokines (IFN-α, IFN-β, IL-6, MIP-1α, and MIP-1β) using ELISA, according to the manufacturer’s instructions. The standard curve was established according to the instructions provided by the kit”. We mainly investigated the effect of overexpression of BPI gene on cytokines. Both the blank group and the NC group served as control groups, and there was no significant difference in the expression of BPI between the two groups (figure 2b). Therefore, we only detected the changes of cytokines in the BPI overexpression group and the NC group.
Lines 117-119 – It is not clear what is the “diluent” in this sentence. Authors should clarify this point by rewriting it.
Response: Thanks for your correction. We have revised the sentence as “Finally, the two E. coli variants were diluted with cell culture medium to obtain a density of 1.0 × 109 colony-forming units (CFU/mL)” (lines 117-118 on page 3).
Lines 122-125 – The counting of the bacterial colonies is not clear. Moreover, it is not clear why authors expressed the result by plotting the number of colonies x 103 instead of a clearer log10 CFU/mL.
Response: Thanks for your correction. We have modified the figure 3.
Line 130 – Why did authors perform this test by incubating cells with E. coli F18 for only 1 hour instead of keeping the incubation time at 2 hours as for the colony count test?
Response: Referring to our previous colony counting test of Dai and Wu et al. [1,2], we optimized the conditions of the experiment according to the different cell lines used in the experiment, and found that 2 hours was better in PK15 cells after performing different time gradient experiments. Therefore, we kept the incubation time at 2 hours in the colony counting test.
- Dai, C.; Yang, L.; Jin, J.; Wang, H.; Wu, S.; Bao, W. Regulation and Molecular Mechanism of TLR5 on Resistance to Escherichia coli F18 in Weaned Piglets. Animals 2019, 27, 735.
- Wu, Z.; Feng, H.; Cao, Y.; Huang, Y.; Dai, C.; Wu, S.; Bao, W. New Insight into the Molecular Mechanism of the FUT2 Regulating Escherichia coli F18 Resistance in Weaned Piglets. Int J Mol Sci 2018, 19, 3301.
Line 132-135 – It is not clear why authors perform a gene expression assay on a DNA sample. It seems more likely a relative quantification of bacteria by evaluating the amount of DNA. Authors should better clarify this part, in particular the result that they show related to the PILIN gene evaluation.
Response: Thanks for your advice. This experiment relied upon the basic principle of relative quantitative PCR to detect and analyze copies of bacterial genes relative to the copies of cellular genes in pigs, in which we used total DNA collected after bacteria adhesion as a template, a cellular gene as an internal gene, and a bacterial gene as a target gene. Because the gene copies were proportional to the number of bacteria or cells, the relative copies of bacterial genes could be equated to the adhesion level of bacteria. Importantly, the relative quantitative detection data could be expressed as the relative adhesion level of bacteria. It is feasible that this experiment could provide insights into the detection of relative amounts of bacterial adhesion to cells [1,2].
- Dai, C.H.; Gan, L.N.; Qin, W.U.; Zi, C.; Zhu, G.Q.; Wu, S.L.; Bao, W.B. Use of Fluorescence Quantitative Polymerase Chain Reaction (PCR) for the Detection of Escherichia coli Adhesion to Pig Intestinal Epithelial Cells. Pol J Vet Sci 2016, 19, 619-625.
- Wu, Z.; Feng, H.; Cao, Y.; Huang, Y.; Dai, C.; Wu, S.; Bao, W. New Insight into the Molecular Mechanism of the FUT2 Regulating Escherichia coli F18 Resistance in Weaned Piglets. Int J Mol Sci 2018, 19, 3301.
Paragraph 2.6 – Authors should clarify which are the bacterial and cellular genes, with their respective housekeeping genes. Moreover, did authors check if the PCR amplification products matched the expected amplicon?
Response: BPI, TLR4, MyD88, CD14, TNF-α, IL-1β, FUT1, and FUT2 genes are cellular genes, and GAPDH is the housekeeping gene for cellular genes. PILIN is a bacterial gene and β-ACTIN acts as a housekeeping gene. Amplification products were confirmed by agarose gel electrophoresis.
Line 143 – Have authors included a purification step for RNA in order to remove all the DNA in the sample?
Response: Thanks for your advice. The DNA removal step in reverse transcription was performed according to the instructions provided by reverse transcription kit (Vazyme Biotech Co., Ltd, Nanjing, China).
Line 148 – Why authors stored cDNA at +4°C, while it is usually stored at -20°C?
Response: We will perform qPCR testing on the same day after reverse transcription, so the cDNA is stored at +4°C. However, prolonged storage needs to be at -20°C.
Lines 153-154 – The protocol of the melting curve is not clear. Authors should revise it.
Response: Thanks for your correction. We have revised the sentence as “In order to analyze the specificity of the amplified products, the melting curve reaction program was carried out at the end of the cycle reaction, and the specific procedure was as follows: 95℃ for 15 s, 60℃ for 60 s, and 95℃ for 15 s” (lines 161-163 on page 4).
Paragraph 2.9 – Authors should add little information about the statistical tests employed in the study to determine statistical significance.
Response: Thanks for your correction. We have revised the sentence as “Data were subjected to ANOVA analysis using SPSS 25.0 software (SPSS, Inc., Chicago, IL, USA), and LSD test was performed for comparison”.
Results
Figure 1 – Authors should comment on the reason why data show a lower BPI expression at 2h. Moreover, authors should also comment on the fact that only E. coli F18ac can significantly increase BPI expression, while F18ab does not exert differences in relative expression levels of the gene.
Response: Thanks for your advice. In the PK15 cell, the expression level of BPI gene was low, and its expression level transiently decreased when initially exposed to LPS, and then began to play a high affinity for the conserved lipid A/inner core of endotoxin, thus inhibiting the proinflammatory activity of LPS [1,2]. With the extension of time, LPS was continuously consumed, and the expression of BPI gene showed a downward trend. We have revised the sentence as “qPCR analysis showed that the stimulation of F18ac strain significantly increased the expression of the BPI gene (P < 0.001), while F18ab strain stimulation had no significant effect on BPI expression (P > 0.05)”.
- Canny, G.; Cario, E.; Lennartsson, A.; Gullberg, U.; Brennan, C.; Levy, O.; Colgan, S.P. Functional and biochemical characterization of epithelial bactericidal/permeability-increasing protein. Am J Physiol Gastrointest Liver Physiol 2006, 290, G557-G567.
- Schultz, H.; Weiss, J.; Carroll, S.F.; Gross, W.L. The endotoxin‐binding bactericidal/permeability‐increasing protein (BPI): a target antigen of autoantibodies. J Leukoc Biol 2001, 69, 505-512.
Lines 174-175 – Authors should also add indications about the significance level of the different letters reported in graph 1a. Is there a P < 0.001?
Response: Thanks for your comments. We have added indications about the significance level of the different letters reported in graph 1a. Different lowercase letters indicate significant difference (P < 0.05), while the same letter means no significant difference (P > 0.05).
Figure 2 – While the pictures of transfected PK15 cells are taken at 96h, it is not clear when the gene expression analysis of BPI has been performed. Moreover, how long is the expression constant and stable? These are all relevant information to allow a complete understanding on the compatibility of the BPI expression with the timeframes selected for the experiments with transfected PK15 cells.
Response: Thanks for your advice. We constructed BPI overexpression lentivirus vector to transfect PK15 cells, and so the BPI gene was stably expressed. The gene expression analysis of BPI was also performed at 96 h.
Figure 5 – In lines 224-226, authors state that the overexpression of the BPI gene can significantly reduce the concentration of several cytokines. In order to confirm this, authors should add a statistical analysis to all the displayed graphs, otherwise the statement cannot be considered true.
Response: Thanks for your advice. We have modified the line chart of figure 5 to a bar chart to more intuitively show the effect of overexpression of BPI gene on the concentration of cytokines. The statistical analysis has been performed on the data displayed in this figure.
Discussion and conclusions
After having addressed all the suggested reviews for the previous parts, authors should revise discussion accordingly, by paying particular attention to the reason why they selected PK15 cells instead of other porcine intestinal cell lines.
Response: Thanks for your comments. We have provided relevant explanations in the discussion section of this manuscript (lines 253-255 on page 9). The key to E. coli infection lies in the expression of E. coli receptor in the porcine intestinal epithelial cells and the difference of innate immunity. At present, the research on E. coli mainly focus on two categories, the screening of E. coli receptors and immune-related genes. The porcine intestinal epithelial cell line is a very suitable in vitro cell model of E. coli receptor genes. This study discusses the effect of porcine BPI gene on E. coli F18 infection and the exploration of immune-relate d functions. The porcine epithelial kidney cell line (PK15) serve as an in vitro cell model for studying immune responses and validating the adhesion pattern of pathogenic E. coli. Based on the function of BPI gene and the aim of this study, we selected PK15 cells as the in vitro cell model for BPI gene in this study.
Moreover, authors should reconsider the statement at lines 251-252 accordingly to the statistical analysis that they should perform on the data shown in figure 5. All the statement and the considerations from line 252 to 259 can be justified and proved only if the statistical analysis shows significant differences between the experimental groups.
The same applies for the final sentence of the conclusions.
Response: Thanks for your comments. The statistical analysis has been performed on the data shown in figure 5 and we have reprocessed this figure. The conclusions were obtained based on the results of statistical analyses.
